# Quantifying exploration in reward-based motor learning

**Nina M. van Mastrigt**[ID]*, **Jeroen B. J. Smeets**[ID], **Katinka van der Kooij**

Department of Human Movement Sciences, Vrije Universiteit Amsterdam, Amsterdam, The Netherlands

* n.m.van.mastrigt@vu.nl

## Abstract

Exploration in reward-based motor learning is observable in experimental data as increased variability. In order to quantify exploration, we compare three methods for estimating other sources of variability: sensorimotor noise. We use a task in which participants could receive stochastic binary reward feedback following a target-directed weight shift. Participants first performed six baseline blocks without feedback, and next twenty blocks alternating with and without feedback. Variability was assessed based on trial-to-trial changes in movement endpoint. We estimated sensorimotor noise by the median squared trial-to-trial change in movement endpoint for trials in which no exploration is expected. We identified three types of such trials: trials in baseline blocks, trials in the blocks without feedback, and rewarded trials in the blocks with feedback. We estimated exploration by the median squared trial-to-trial change following non-rewarded trials minus sensorimotor noise. As expected, variability was larger following non-rewarded trials than following rewarded trials. This indicates that our reward-based weight-shifting task successfully induced exploration. Most importantly, our three estimates of sensorimotor noise differed: the estimate based on rewarded trials was significantly lower than the estimates based on the two types of trials without feedback. Consequently, the estimates of exploration also differed. We conclude that the quantification of exploration depends critically on the type of trials used to estimate sensorimotor noise. We recommend the use of variability following rewarded trials.

## Introduction

Imagine you are in a crowded train and all seats are occupied. On the brake at the first stop, you almost fall over. To prevent a reoccurrence on the following stops, you try to find a way to change your posture during the brake. This exploration results in a variety of posture shifts. Once you have found a way that prevents you from falling, you stop exploring and try to reproduce that shift of posture on the next stop to successfully keep your balance. However, an exact reproduction is impossible because both the planning and execution of your movements are inherently noisy [1–3]. Variability in your movements can thus be caused by a combination of exploration and sensorimotor noise. The question we address in this paper is: how can we experimentally separate exploration from sensorimotor noise?

**Data Availability Statement:** All center-of-pressure signal files, questionnaire data and motivation questionnaire data are available in the Open Science Foundation repository (https://osf.io/x7hp9/).

**Funding:** The research was funded by the Nederlandse Organisatie voor Wetenschappelijk Onderzoek, Toegepaste en Technische Wetenschappen (NWO-TTW), by the Open Technologie Programma (OTP) grant 15989 awarded to Jeroen Smeets.

**Competing interests:** The authors have declared that no competing interests exist.

Separating exploration from sensorimotor noise is especially relevant when studying learning from binary reward feedback, i.e. 'reward-based' motor learning [1,4–6]. Binary reward feedback does not provide information on error size or direction, as with binary hit-miss information in goal-directed movements. If you want to improve your performance but receive no feedback on error size and direction, you have to explore which actions lead to reward such that rewarded actions can be exploited [1,7]. Indeed, studies into reward-based motor learning have shown a higher variability in motor output following non-rewarded movements than following rewarded movements [4,5,8–14]. Moreover, there are indications that initial variability in motor output, regarded as exploration, is positively related with learning, both in songbirds [15] and in humans [16]. Yet, a limited amount of evidence exists for a relationship between exploration and motor learning [1,17,18]. This may relate to the fact that exploration is difficult to measure because variability consists of multiple sources [1,17]. Examples are planning noise, execution noise and perceptual noise [1,17], which we summarize as sensorimotor noise and exploration [11,14,19].

Although the concept 'exploration' is frequently used in the literature on reward-based motor learning, the concept is ill-defined. Consequently, various measures have been used to quantify exploration. Wu and colleagues [16] considered the variability in a baseline phase as exploration. Others considered variability in the presence of feedback [4,5,8–10,13] as exploration. We want to limit the concept of exploration to the variability that can be used in motor learning. This exploration is expected to be present when someone perceives opportunity for learning, i.e. in the presence of feedback. In that case, variability will include exploration, especially following non-rewarded trials [4,5,8,10,13,20]. However, this variability will additionally include sensorimotor noise. To exclusively quantify exploration, we thus need an estimate of sensorimotor noise.

In a reaching task, van der Kooij and Smeets [9] estimated sensorimotor noise as the median trial-to-trial change in baseline trials without feedback, assuming that participants have no reason to explore when no reward feedback is available. To quantify exploration, they subtracted this noise estimate from total variability in feedback trials. However, for some participants variability appeared smaller in the feedback trials than in the baseline. Theoretically, this is not expected, because exploration should *increase* variability. This shows that there was a problem in quantifying sensorimotor noise. The quantification may have been unreliable or may have been invalid. The latter might have been the case because the estimates of sensorimotor noise alone and in combination with exploration were obtained in different feedback contexts, i.e. in trials without feedback versus trials with feedback. Sensorimotor noise has been found to depend on the opportunity to obtain reward [21–23]. In addition, feedback context may influence variability through a motivational effect [24]. This raises the question in which feedback context sensorimotor noise can best be quantified. An additional problem is the choice of summary statistics to describe variability [25,26], especially when extreme values are present that may either represent exploration or outliers.

In summary, estimating the level of sensorimotor noise is essential in unravelling the relationship between exploration and reward-based learning [11,17,19]. We used a reward-based weight-shifting task to compare different methods for quantifying exploration. To eliminate the influence of learning on variability, feedback was stochastic. To verify whether this task results in exploration, we first aimed to replicate the finding that variability following non-rewarded trials is higher than variability following rewarded trials [4,5,8–14]. Next, we aimed to compare the estimation of sensorimotor noise in three feedback contexts: (1) trials in a baseline phase without feedback, (2) trials in short blocks without feedback that were flanked by short blocks of feedback trials and (3) rewarded trials in blocks of feedback trials. We also aimed to explore the statistical assessment of sensorimotor noise and exploration: which

summary statistic should be used, how does variability change over time and how uncertain are sensorimotor noise and exploration estimates? We found that the quantification of exploration depended critically on the feedback context used to estimate sensorimotor noise.

## Methods

### Participants

Forty-four healthy adults participated in the experiment (14 male, 30 female). They were recruited either from the personal network of the first author or at the Faculty of Behavioural and Movement Sciences of the Vrije Universiteit Amsterdam. Psychology students (N = 13) received course credits for their participation, while other participants received no compensation. Participants were unaware of the research aim and participated voluntarily. All participants provided written informed consent prior to participation. They reported that they were able to stand independently for at least 30 minutes and had no lower body injuries preventing them from shifting weight during stance. In addition, they reported no visual, auditive, balance and cognitive deficits.

Participants were aged between 18 and 59 years old. Of the initial 44 participants, 41 were included in the data analysis (27±7 years; 14 male, 11 non-Dutch speaking). One participant was excluded due to a missing data file, and two participants were excluded because they reported having noticed the stochasticity of the feedback in the exit interview. The study was approved by the Scientific and Ethical Review Board (VCWE) of the Faculty of Behavioural and Movement Sciences of the Vrije Universiteit Amsterdam (approval number 2019–104).

### Set-up

Participants stood barefoot on a custom-made 1x1 m eight-sensor strain gauge force plate (sampling frequency: 100 Hz) that was used to measure center-of-pressure position [27]. The dorsal heel borders were positioned 9 cm behind the center of the force plate and the medial borders 18 cm apart (Fig 1A). At eyelevel, 1.5 m in front of the participant, a 55-inch monitor displayed a schematic representation of the feet, center and target, a short instruction, trial number and progress bar (Fig 1A and 1B). To synchronize the visual stimulus timing with the force plate signal, we recorded the signal of a photodiode that was attached to the monitor with the force plate data.

### Task

Participants performed a repetitive weight-shifting task. On each trial, their task was to hit with their center-of-pressure a fixed target within their base of support. On the screen, the target was visually represented as a circle with a diameter of 0.5 cm, positioned 1 cm in front of and 5 cm to the left of a center cross. Participants were instructed to keep their feet fixed to the ground, to lean towards the target, and once they hit it, immediately start moving back to neutral stance, within 1.5 s after the start beep (Fig 1B).

During this 1.5 s period, an instruction screen with a schematic representation of the feet, center and target was displayed, along with a short instruction, trial number and progress bar (Fig 1B). The screen did not display center-of-pressure position. After 1.5 s, participants received either no performance feedback (non-feedback trials), or non-veridical binary reward feedback (feedback trials) for 0.5 s. After non-feedback trials, a neutral beep marked the end of the trial and "Score hidden" was displayed in white on a black screen. After feedback trials, a black screen with the current total score was presented. Reward consisted of a bell sound to mark the end of the trial, five points added to the score, and the total score coloring green.

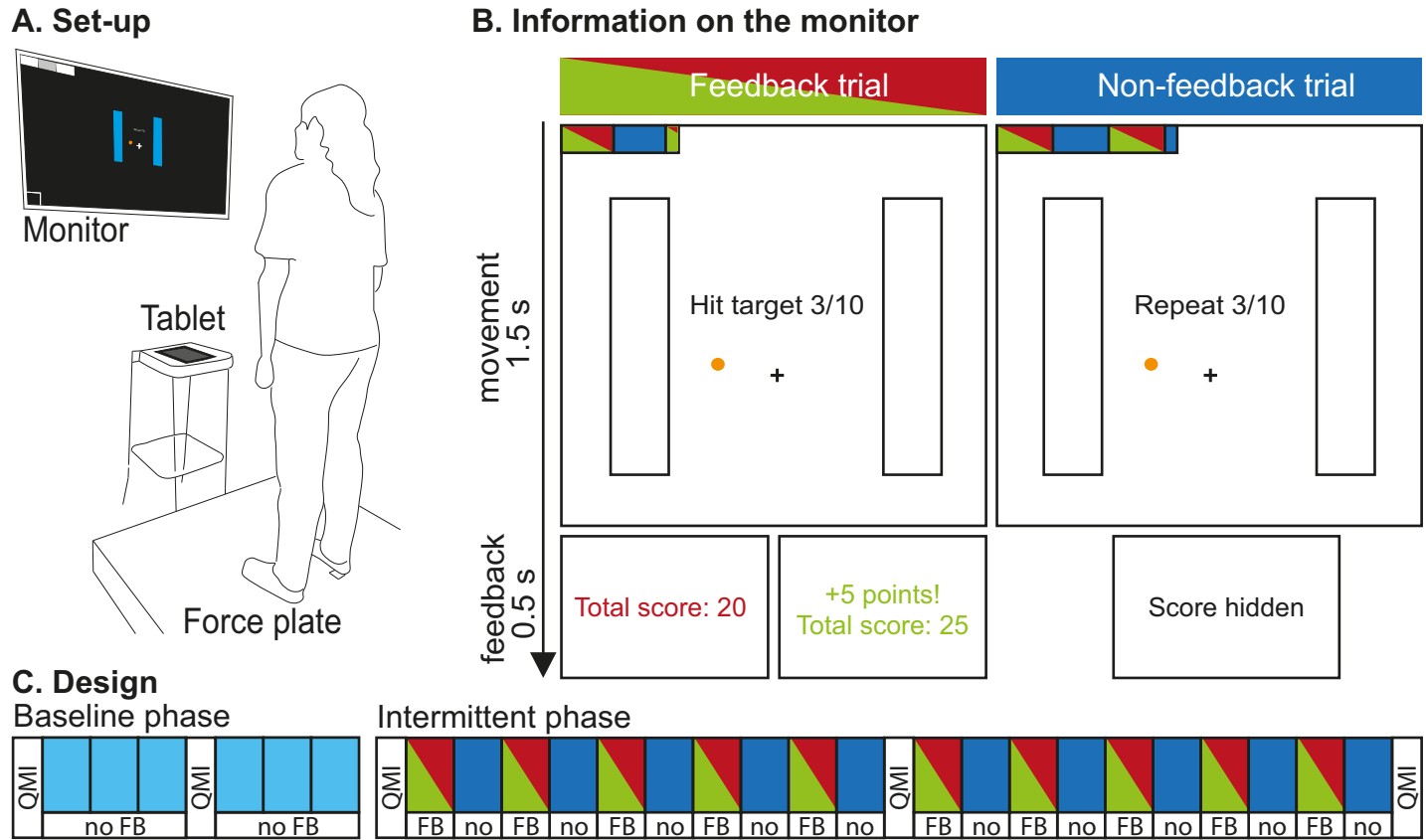

**Fig 1. Experimental set-up and design.** (A) Overview of the set-up. (B) Information as presented on the monitor. At the onset of and during a trial, the screens displayed a graphical instruction (upper panels). Vertical bars represented the foot position as instructed, the cross represented the center from which to start, and the orange circle represented the target. The horizontal bar in the top left corner represented the progress bar. Instruction strings during the movement depended on trial type. Stochastic binary score feedback was provided following feedback trials, and "Score hidden" was displayed following non-feedback trials (lower panels). (C) Design. A baseline phase was followed by an intermittent phase. Each block contained 10 trials, except for the last block which contained 11 trials. Green-and-red blocks indicate feedback trials; blue blocks indicate non-feedback trials. White bars indicate motivation assessments.

Reward absence consisted of a negative buzzer and the current total score coloring red. Although the instruction stated that each target hit would be rewarded with five points, the reward feedback that participants received was independent of their performance: reward was provided in half of the feedback trials, distributed in random order. The feedback was followed by an empty screen depicted for 1 s, to allow participants to start the next trial from a neutral stance.

## Procedure

Participants were told that the experiment was about the control of their center-of-pressure and how they learned from reward feedback. They were informed that the procedure would last maximally 30 minutes. After signing the informed consent form, they received verbal and visual instructions on the task. Participants were told that each hit would be rewarded by five points, both in feedback and non-feedback trials, but that the interim score would only be visible in feedback trials. They were informed that after a familiarization phase, they would perform two phases: first a baseline phase containing non-feedback trials only, and second an intermittent phase containing blocks with or without feedback (Fig 1C). The experimenter told participants that the sum of the two end scores would be attached to a leader board on the wall.

Before the actual experiment, participants could familiarize themselves with the set-up. They received on-line visual center-of-pressure feedback to clarify the relation between their changes in posture and the center-of-pressure movement as indicated on the screen. This familiarization consisted of one 30 s period in which participants were encouraged to find the boundaries of their base of support, one sequence of four trials with different targets, and one sequence of five trials with the experimental target. After the familiarization, the on-line visual center-of-pressure feedback disappeared. Subsequently, participants received auditory and visual instructions on the experimental procedure, with separate instructions for the baseline and intermittent phase. After these instructions, participants practiced 23 trials of the intermittent phase, to practice timing of their movements and switching between feedback and non-feedback trials.

All participants performed the baseline phase first (Fig 1C). This phase consisted of 61 non-feedback trials and was designed to obtain a first estimate of sensorimotor noise. Secondly, participants performed an intermittent phase. This phase consisted of 201 trials, with blocks of 10 trials with stochastic reward feedback (feedback trials), and 10 trials without performance feedback (non-feedback trials). Non-feedback blocks were used to obtain a second estimate of sensorimotor noise, and feedback blocks were used to obtain a third estimate of sensorimotor noise based on rewarded trials, and to quantify exploration in non-rewarded trials. For the non-feedback trials, the instruction for participants was to repeat the movement to where they thought the target was at that moment. During these trials "Repeat" was displayed at the center of the screen (Fig 1B). For the feedback trials, the instruction for participants was to hit the target using the reward feedback they received, and during these trials, "Hit target" was displayed at the center of the screen (Fig 1B). Participants performed about 20 trials per minute, so the baseline phase lasted about 3 minutes and the intermittent phase 10 minutes.

Variability might increase due to loss of motivation. Therefore, we assessed the motivation of our participants at the start, halfway point and end of each phase. We used the Quick Motivation Index (QMI) [28] to assess (1) how much the participant had enjoyed the task until that moment and (2) how motivated the participant was to continue (Fig 1A–1C). Both questions were answered on a continuous scale with extremes "Not at all" and "Very much", using a slider on a tablet (Apple iPad 2, screen diameter 10 inch). After the weight-shifting task, participants filled in a questionnaire with 15 statements about their movement strategies, cognitive involvement, feedback, motivation and fatigue. These data were collected to be able to check anomalous findings in the movement data. Finally, the experimenter debriefed the participant verbally and checked whether participants were aware of the feedback manipulation. The participants who were aware that the feedback was unrelated to their performance were excluded because they might not have related their exploration to the feedback.

## Data analysis

**Trial-to-trial analysis of variability.** We assessed variability based on changes in movement endpoint from trial to trial. We filtered the force plate signal using a 5-point moving average [27]. We defined a movement endpoint as the most radial center-of-pressure position within a trial. As movement endpoints can drift, even when providing stochastic feedback [8,14], we did not analyze variability based on the distribution of endpoints around their mean, which is described by the standard deviation and variance of this distribution. Instead, we analyzed variability based on the distances between movement endpoints of two subsequent trials: the trial-to-trial changes in movement endpoints. If movement endpoints are normally distributed with a variance $\sigma^2$, the relation between mean trial-to-trial change ($\Delta$) and the variance is given by $\Delta^2 = \pi \sigma^2$ [29].

We assumed that two sources of variability contributed independently to the total variability that we observed in movement endpoints: sensorimotor noise and exploration. If two sources of variability are independent, the variance of the combined measure equals the sum of the variances of the two constituents [30]. If we assume that each movement endpoint is drawn from a two-dimensional normal distribution, the mean trial-to-trial change $\Delta$ is proportional to the standard deviation $\sigma$ of that distribution ($\Delta = \sigma\sqrt{\pi}$) [29]. We can thus formalize the addition of the two components of variability in terms of mean trial-to-trial changes as:

$$\Delta^2 = \Delta_m{}^2 + \Delta_\eta{}^2 \tag{1}$$

In this equation, $\Delta_m{}^2$ represents the trial-to-trial changes caused by pure sensorimotor noise and $\Delta_\eta{}^2$ represents the trial-to-trial changes due to exploration only. We approximated $\Delta$ by the median of trial-to-trial changes, instead of the mean, since squared trial-to-trial change distributions are positively skewed [29].

We estimated sensorimotor noise by assuming that following non-feedback and rewarded trials, participants have no reason to explore [9]. In this way, we could obtain sensorimotor noise estimates in three feedback contexts: following non-feedback trials in the baseline phase, following non-feedback trials in the intermittent phase, and following rewarded trials in the feedback blocks of the intermittent phase. We used the last 51 trials of the baseline phase to calculate 50 trial-to-trial changes following non-feedback trials. The first 10 trials were discarded because participants may have needed some time before finding a consistent way of performing the task. We used all 201 trials of the intermittent phase to calculate 200 trial-to-trial changes, of which 100 were trial-to-trial changes following non-feedback trials and 50 were trial-to-trial changes following rewarded trials.

We assumed that participants start exploring after having failed to obtain reward feedback. We used all 50 trial-to-trial changes following non-rewarded trials to calculate variability that included exploration. We could quantify exploration ($\Delta_\eta{}^2$) by subtracting sensorimotor noise, as defined by $\Delta^2$ following non-feedback or rewarded trials, from $\Delta^2$ following non-rewarded trials, according to Eq 1.

**Uncertainty in the estimate of variability.** To assess the uncertainty in variability estimates for each participant, we calculated a 95% confidence interval of 10.000 bootstrap estimates of the median of randomly selected squared trial-to-trial changes ($\Delta^2$) with replacement. For each bootstrap estimate of $\Delta^2$ following non-feedback trials in the baseline or intermittent phase, 50 or 100 trial-to-trial changes were available for random selection, respectively (Fig 1C). For each bootstrap estimate of $\Delta^2$ following rewarded or non-rewarded trials, 50 trial-to-trial changes were available for random selection (Fig 1C). To assess the uncertainty in exploration estimates, we calculated the 95% confidence interval of 10.000 estimates of $\Delta_\eta{}^2$ obtained by subtracting 10.000 bootstrap estimates of sensorimotor noise from $\Delta^2$ following non-rewarded trials, according to Eq 1. To assess how the uncertainty in variability estimates depends on the number of trials, we performed the bootstrap analysis with 5 to 50 randomly selected trials.

To assess consistency of variability estimates over time, we analyzed variability in each block of ten trials, with $\Delta^2$ calculated as the median of squared trial-to-trial changes of the ten trials in a block (Fig 3B). We also assessed uncertainty in variability estimates based on this blocked analysis: the median variability estimate of the individual blocks. Now, for each bootstrap estimate of $\Delta^2$, 5 or 10 block estimates of $\Delta^2$ were available for random selection with replacement. Due to the random reward sequence, not every estimate of $\Delta^2$ following rewarded or non-rewarded trials was based on the same number of trials. Therefore, the probability of randomly selecting an estimate of $\Delta^2$ following rewarded or non-rewarded trials for the

bootstrap analysis was set to correspond to the amount of rewarded or non-rewarded trials in that block.

**Motivation and enjoyment.** For each participant the ratings on the two questions in the QMI were averaged [28]. This way we obtained five motivation scores: pre-baseline phase, baseline phase, pre-intermittent phase, intermittent phase and post-intermittent phase.

### Statistics

Within-participant distributions of trial-to-trial changes and between-participant distributions of variability were positively skewed (Fig 2A). Therefore, we tested our hypotheses with non-parametric tests and reported medians and interquartile ranges. To check whether our task

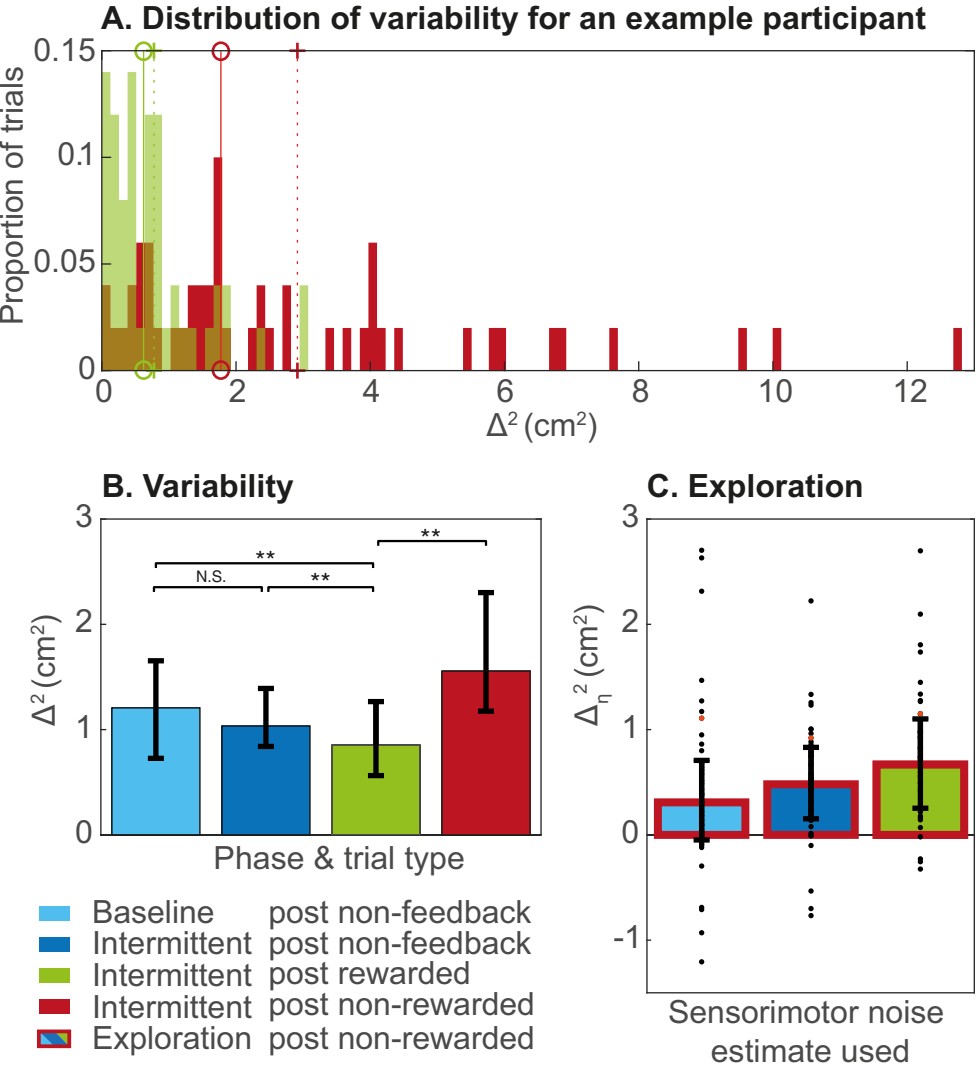

**Fig 2. Variability and exploration.** The color of the bar indicates which trials are used to estimate variability. (A) Distribution of squared trial-to-trial changes following rewarded (green) and non-rewarded (red) trials for an example participant. Vertical lines indicate the median (solid) and mean (dotted) of the distribution. (B) Variability ($\Delta^2$) on all trial types. Bars indicate the median across participants and error bars the interquartile range. Double stars indicate significant differences at p<0.01. (C) Exploration ($\Delta_\eta^2$) following non-rewarded trials, calculated based on Eq 1 plotted in the same format as panel A. Bars indicate the median across participants and error bars the interquartile range. Dots indicate the data of individual participants; the red dot indicates the example participant of panel A.

induced exploration, we tested whether variability following non-rewarded trials would be higher than following rewarded trials using a two-sided Wilcoxon signed rank test. To test whether sensorimotor noise estimates depended on feedback context, we used a Friedman's ANOVA with post-hoc Wilcoxon signed-rank tests corrected for multiple comparisons. We did not test explicitly whether exploration estimates depended on feedback context in which sensorimotor noise was estimated. The reason for this is that the dependence of exploration on feedback context is the same as that of sensorimotor noise: within a participant, the various estimates of exploration are based on the same value of the variability following non-rewarded trials.

As motivation may influence variability, we tested whether motivation depended on feedback context by comparing the change in motivation during the baseline and intermittent phase using a two-sided Wilcoxon signed-rank test, and whether this resulted in a different motivation halfway through each phase with Wilcoxon signed-rank tests. Furthermore, we calculated the Spearman's correlation between changes in motivation and sensorimotor noise from baseline to intermittent test.

## Results

We visually examined the distributions of squared trial-to-trial changes underlying the variability as quantified by the summary statistic we use in our between-participant analyses ($\Delta^2$) (Fig 2A). Squared trial-to-trial changes displayed a positively skewed distribution with a long right tail, as shown in Fig 2A. This was the case for rewarded as well as for non-rewarded trials. For the majority of participants, the distribution of trial-to-trial changes following non-rewarded trials showed a longer tail than the distribution of trial-to-trial changes following rewarded trials (e.g. Fig 2A). Due to the rightward skew, mean squared trial-to-trial changes were considerably higher than median squared trial-to-trial changes ($\Delta^2$) (Fig 2A). Our data thus supported the use of the median of squared trial-to-trial changes as a summary statistic for within-participant variability.

In a first statistical test, we checked whether the task elicits exploration following non-rewarded trials. This was indeed the case: a Wilcoxon signed-rank test confirmed that $\Delta^2$ following non-rewarded trials (Mdn = 1.6 cm$^2$) was significantly higher than $\Delta^2$ following rewarded trials (Mdn = 0.8 cm$^2$), z = -5.15, p < 0.001, r = -0.57 (Fig 2A).

The main statistical test assessed whether sensorimotor noise estimates depended on feedback context. A Friedman's ANOVA on the $\Delta^2$ following baseline non-feedback trials, following intermittent non-feedback trials, and following rewarded trials in the intermittent phase, revealed a significant influence of feedback context on variability, $\chi^2$ (2) = 10.68, p = 0.005 (Fig 2B). Post-hoc Wilcoxon signed-rank tests indicated that $\Delta^2$ following rewarded trials (Mdn = 0.8 cm$^2$) was significantly lower than $\Delta^2$ following baseline non-feedback trials (Mdn = 1.1 cm$^2$), z = -2.64, p = 0.008, r = -0.29, and $\Delta^2$ following intermittent non-feedback trials (Mdn = 1.0 cm$^2$), z = -3.22, p = 0.001, r = -0.36 (Fig 2B). The $\Delta^2$ following baseline non-feedback trials (Mdn = 1.2 cm$^2$) and $\Delta^2$ following intermittent non-feedback trials (Mdn = 1.0 cm$^2$) did not differ significantly, z = -0.24, p = 0.81 (Fig 2B).

Using the three estimates of sensorimotor noise, we calculated exploration ($\Delta_\eta^2$) according to Eq 1. Fig 2C shows the median $\Delta_\eta^2$ across participants, along with interquartile range and individual data. Since sensorimotor noise estimates differed significantly, the corresponding estimates of $\Delta_\eta^2$ also differed significantly. Calculating $\Delta_\eta^2$ based on sensorimotor noise following rewarded trials resulted in the highest value for the median exploration over participants and the least participants with negative exploration values. No statistical tests were conducted.

In Fig 3, we provide some data to illustrate the reliability of our estimate of sensorimotor noise and exploration. The uncertainty in our estimates of sensorimotor noise and exploration

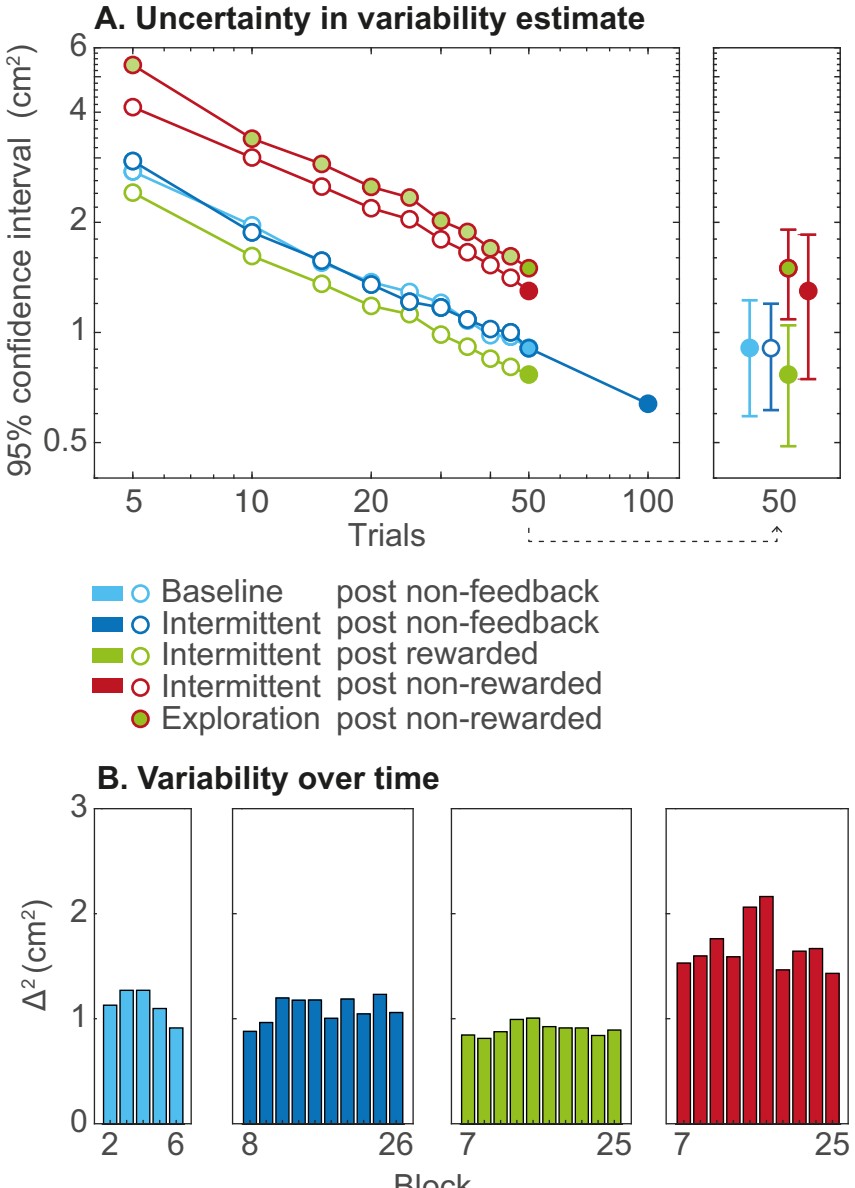

**Fig 3. Uncertainty in the estimates of variability.** (A) Uncertainty of variability ($\Delta^2$) as a function of number of trials based on which it was estimated (log-log plot). Circles indicate the median across participants. Filled circles correspond to the variability and exploration as presented in Fig 2B and 2C. The right panel provides an indication of variations across participants of this uncertainty: the error bars indicate the interquartile range across participants. (B) Variability estimates ($\Delta^2$) per non-feedback block of 10 trials in the baseline phase and in the intermittent phase, and $\Delta^2$ estimates following rewarded and non-rewarded trials per feedback block in the intermittent phase. Bars indicate the median across participants.

based on 50 trials was comparable to the estimates themselves (compare Fig 2B and 2C with Fig 3A, uncertainty based on 50 trials). The uncertainty in the estimates of sensorimotor noise and exploration decreases with number of trials to estimate variability. The slope of the lines in Fig 3A is -0.5, which corresponds to an uncertainty that is proportional to $1/\sqrt{N_{\text{trials}}}$. Extrapolation shows how our experiment could have benefited from a higher number of trials. The uncertainty of our estimates of variability following non-feedback and rewarded trials was

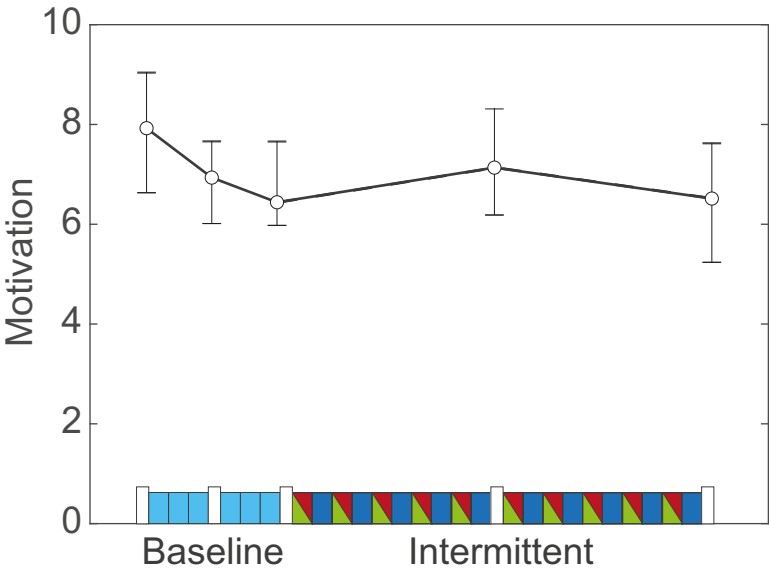

**Fig 4. Motivation.** Median of motivation scores across participants over time. Error bars indicate interquartile range. The blocks along the x-axis indicate the feedback context in the same format as Fig 1C. Motivation decreases during the baseline phase, increases during the first half of the intermittent phase and decreases again in the second half of the intermittent phase.

lowest. As the estimate of exploration ($\Delta_\eta^2$) is based on the difference between two measures of variability, its uncertainty was higher than the uncertainty in either of them.

We visually examined trends in variability over blocks. No marked trends in variability over blocks were observed (Fig 3B). We furthermore checked whether considering the order of blocks would influence the uncertainty. The uncertainty of variability estimated based on block estimates was in the same order of magnitude (not shown; median 95% confidence interval sizes of 1.1, 0.6, 0.8 and 1.4 cm$^2$). Uncertainty in the quantification of exploration based on the three feedback contexts based on the individual blocks was also comparable to the uncertainty determined irrespective of the blocks (not shown; 2.4, 1.9 and 1.9 cm$^2$).

As expected, feedback context influenced motivation (Fig 4): during the baseline phase motivation decreased (Mdn = -0.55), whereas during the intermittent phase it increased (Mdn = 0.49), $z$ = -3.22, $p$ = 0.001, $r$ = -0.37. Nevertheless, motivation scores halfway the baseline phase (Mdn = 7.0) and halfway the intermittent phase (Mdn = 7.2) did not differ significantly, $z$ = -1.23, $p$ = 0.22. Furthermore, changes in motivation from baseline to intermittent phase were not significantly correlated with changes in sensorimotor noise between the baseline and intermittent phase, $\rho$ = 0.03, $p$ = 0.88.

## Discussion

We used a weight-shifting task with stochastic binary reward feedback to compare methods for quantifying exploration in reward-based motor learning. Participants performed a baseline phase with non-feedback trials, followed by an intermittent feedback phase consisting of alternating blocks with or without feedback. In the feedback blocks, participants exhibited higher variability following non-rewarded than following rewarded trials, which we regard a signature of exploration. We found higher sensorimotor noise in the blocks without feedback than based on the variability following rewarded trials. The sensorimotor noise estimates in the non-feedback blocks did not differ between the baseline and the intermittent phase, neither did

motivation differ between the two phases. Our estimates of sensorimotor noise were stable, but the uncertainty in the estimates of sensorimotor noise and exploration was in the same order of magnitude as the estimates themselves. Therefore, sensorimotor noise can best be estimated using the median trial-to-trial change following rewarded trials, based on a large number of trials.

Variability following rewarded trials was lower than the variability in the blocks without feedback (compare red and green bars in Fig 2A and 2B). This can be interpreted in two ways. A first interpretation is that some exploration occurs in blocks without feedback, despite the instruction to repeat their movements as precisely as possible. A second interpretation for the higher variability in blocks without feedback is that sensorimotor noise is larger than with feedback. The lack of feedback may have resulted in boredom and participants ignoring the instructions to repeat their movements a precisely as possible. Additionally, sensorimotor noise may have been downregulated in anticipation of reward on feedback trials. This interpretation is in line with the results of Manohar and colleagues [21], who showed that motor noise could be decreased in anticipation of reward in an eye-movement task with auditory maximum reward cues. Both interpretations of the difference in sensorimotor noise estimates lead us to the conclusion that variability following rewarded trials yields the best sensorimotor noise estimate for quantifying exploration.

An unexpected finding was the high uncertainty of our estimates of variability (Fig 3A). Based on our 50–100 trials, our estimates of variability and exploration were very uncertain. This finding implies that studying sensorimotor noise and exploration requires a high number of trials. In the literature, human motor learning experiments usually seem to contain no more than 200 feedback trials [8,10,12,31–36], although some studies report higher numbers, up to 900 trials [4,5,11,16,19,37,38]. An example of a study incorporating many trials is [14], in which rats performed 300k trials on average. Importantly, this study also found that variability following rewarded trials is lower than following non-rewarded trials. Based on their finding that variability decreases with higher reward rates, they consider the minimum variability obtained with maximum reward rates as "unregulated variability" to which "regulated variability" is added if no reward is received. We used a similar approach, with unregulated variability replaced by sensorimotor noise and regulated variability by exploration. To use such a large number of trials in humans is often not feasible. A positive implication of our results is that no baseline trials have to be included specifically for assessing sensorimotor noise, since exploration can best be estimated in blocks with feedback, based on trial-to-trial changes following reward.

We quantified exploration using the assumption that the observed variability following non-rewarded trials is the sum of the variabilities of sensorimotor noise and exploration (Eq 1). Exploration ($\Delta_\eta^2$) is by definition positive. However, even when using the lowest sensorimotor noise estimate, we obtained negative estimates of $\Delta_\eta^2$ for 3 of the 41 participants. A similar problem was present in [6,9]. The most obvious explanation for these negative estimates is the high uncertainty in our statistical assessment. If one finds negative estimates of $\Delta_\eta^2$ with a much higher number of trials, this might imply that sensorimotor noise and exploration are not independent, contrary to what is commonly assumed [6,11,14,19]. For instance, sensorimotor noise may increase when exploring, because reward is less certain when exploring [21]. Alternatively, sensorimotor noise may be downregulated to increase opportunity for learning [11].

In the introduction, we proposed that feedback context may influence variability through a motivational effect. We found no difference in motivation between the baseline and intermittent phase, and accordingly no difference in sensorimotor noise estimates in the corresponding non-feedback trials. This might be explained by the fixed order of the phases. A possible

decrease in motivation over time as has been observed in [28] may have been counteracted by the motivational effect of feedback in the intermittent test. The different change in motivation during the first half of the baseline phase and the first half of the intermittent phase supports this conclusion: motivation decreased during the baseline phase, whereas it increased during the intermittent phase.

We observed no trends in variability over time, neither in the amount of exploration nor in our sensorimotor noise estimates. One might have expected an increase in variability due to loss of motivation, but the changes in motivation we observed (Fig 4) were small. Alternatively, one might have expected a decrease in variability over time due to learning [1,5], but in our experiment the use of stochastic binary reward feedback effectively prevented participants from learning.

We quantified exploration using the median as a summary statistic for variability. We recommend doing so when calculating exploration based on absolute changes, because this indeed yields a positively skewed distribution (Fig 2A). The median is less sensitive to extreme values than the mean. As such extreme values were indeed present (Fig 2A), the median better reflects the central tendency of the distribution of squared trial-to-trial changes. We therefore consider the median the most suitable summary statistic for our method.

Using a weight-shifting task, we replicated a finding from studies using arm movements: increased variability following non-rewarded trials as compared to rewarded trials [4,5,8–13]. In addition, we found a ratio between variability due to exploration and sensorimotor noise of 0.78. This ratio is similar to the ratio of about 0.6 for young healthy adults reported by Therrien and colleagues [19] and 1.2 for healthy adults reported by Cashaback and colleagues [6], which they obtained by fitting a model including exploration and sensorimotor noise as independent sources of variability. Our study may thus indicate that results of previous reward-based motor learning studies can be generalized to tasks involving other limbs. Our study however also yields some new questions. Firstly, the amount of trials necessary to reliably quantify exploration should be studied. Secondly, an important question is whether sensorimotor noise and exploration are independent sources of variability, especially since sensorimotor noise may be up- or downregulated following non-rewarded trials.

To conclude, we showed that the effect of reward feedback on variability can be induced in a target-directed weight-shifting task with stochastic reward feedback. The results of the current study have two important methodological implications for those who study exploration in reward-based motor learning. First, exploration can best be quantified using sensorimotor noise estimated from trial-to-trial changes following rewarded trials. Second, since sensorimotor noise and exploration represent sources of variability, a much larger number of trials should be used to quantify these sources than is commonly done.

## Author Contributions

**Conceptualization:** Nina M. van Mastrigt, Jeroen B. J. Smeets, Katinka van der Kooij.

**Funding acquisition:** Jeroen B. J. Smeets, Katinka van der Kooij.

**Investigation:** Nina M. van Mastrigt.

**Methodology:** Nina M. van Mastrigt, Jeroen B. J. Smeets, Katinka van der Kooij.

**Software:** Nina M. van Mastrigt.

**Supervision:** Jeroen B. J. Smeets, Katinka van der Kooij.

**Visualization:** Nina M. van Mastrigt, Jeroen B. J. Smeets, Katinka van der Kooij.

**Writing – original draft:** Nina M. van Mastrigt.

**Writing – review & editing:** Nina M. van Mastrigt, Jeroen B. J. Smeets, Katinka van der Kooij.

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
