## [Decision Letter · Decision Letter 0]

3 Jan 2020

PONE-D-19-33458

Quantifying exploration in reward-based motor learning

PLOS ONE

Dear Mrs. van Mastrigt,

Thank you for submitting your manuscript to PLOS ONE. After careful consideration, we feel that it has merit but does not fully meet PLOS ONE’s publication criteria as it currently stands. Therefore, we invite you to submit a revised version of the manuscript that addresses the points raised during the review process.

I was able to obtain one review from an expert in the field.  Overall, this reviewer had a positive impression of the paper, but also noted some issues that should be addressed in a revision.  From my reading of the manuscript I agree with the reviewer, and I invite you to submit a revision.  If you choose to submit a revision I will likely ask the same reviewer to review the revised manuscript.

We would appreciate receiving your revised manuscript by Feb 17 2020 11:59PM. To enhance the reproducibility of your results, we recommend that if applicable you deposit your laboratory protocols in protocols.io, where a protocol can be assigned its own identifier (DOI) such that it can be cited independently in the future. For instructions see: http://journals.plos.org/plosone/s/submission-guidelines#loc-laboratory-protocols

We look forward to receiving your revised manuscript.

Kind regards,

Darrell A. Worthy, Ph.D

Academic Editor

PLOS ONE

Journal Requirements:

2. Please provide additional details regarding participant consent. In the Methods section, please ensure that you have specified (1) whether consent was informed and (2) what type you obtained (for instance, written or verbal). If your study included minors, state whether you obtained consent from parents or guardians. If the need for consent was waived by the ethics committee, please include this information.

Reviewers' comments:

Reviewer's Responses to Questions

**Comments to the Author**

1. Is the manuscript technically sound, and do the data support the conclusions?

Reviewer #1: Yes

2. Has the statistical analysis been performed appropriately and rigorously? 

Reviewer #1: Yes

3. Have the authors made all data underlying the findings in their manuscript fully available?

Reviewer #1: Yes

4. Is the manuscript presented in an intelligible fashion and written in standard English?

Reviewer #1: Yes

5. Review Comments to the Author

Reviewer #1: In this study the authors systematically investigate ways of quantifying exploration in motor learning by separating it from variability due to sensorimotor noise. Using a target directed weight shift task, they designed an experiment wherein multiple different baselines for calculating variability due to exploration (which have all been variously used in the literature) can be used and compared. They conclude that using trial-to-trial change following rewarded trials provided a better baseline than trials in a baseline block or trials in no-feedback blocks. They also conclude (based on assessment of the uncertainty in the estimates) that the typical number of trials used may be insufficient for accurate estimations of variability due to exploration.

This is a nice study with a clever design that provides a valuable contribution that can help improve methods for future studies. The methods and analyses seem appropriate and sound. The conclusion seems justified and makes theoretical sense to me. Quantifying the uncertainty in the estimations was a nice touch. I think that it should be accepted for publication, but I think there are some issues the authors can address first.

--

How long did the whole experiment take participants approximately? This should be stated in the methods section somewhere. It is particularly important since you recommend an order of magnitude greater number of trials to be used in the future, and it would be good to have an idea of how long an experiment with that many trials would take. Would it be unreasonably long, or long enough that the participants’ motivation would become an issue?

Here is one suggestion: you could expand on the bootstrapping analysis uses to quantify uncertainty by doing some simulations with larger numbers of trials and seeing how large the uncertainty of the estimate is. For example, there could be a graph of uncertainty as a function of number of trials. This could be really useful in determining the minimum number of trials to get a decent estimate. It might be that fewer additional trials are needed than your claim. At the other extreme, it’s possible that it asymptotes and even an unreasonably large number of trials gives a poor estimate. It would be cool (and potentially useful) to see what the curve of uncertainty by number of trials looks like.

Pg. 14 “No marked trends in variability over blocks were observed (Fig 3C).” I think that this point could use a little more discussion. I guess that it is good for the goals of the study that variability was constant across blocks, but shouldn’t you expect some decrease over time? Learning should decrease variability, and in general exploration should be highest early in a task when there is more to learn and should taper off later. Is it that most of the learning happens in the familiarization phase and beginning of the baseline block? Or is it that the stochastic feedback effectively prevents any real learning from occurring? A little discussion of these issues could be helpful. I think it makes sense that variability was stable, but my initial expectation was that it would go down over time.

At the end of the introduction (pg. 4) you say “We also aimed to explore the statistical assessment of sensorimotor noise and exploration: which summary statistic should be used, how does variability change over time and how uncertain are sensorimotor noise and exploration estimates?” It would be nice to return to these questions a little bit more directly in the Discussion. The last question is discussed in detail, but the other two are only briefly touched on. Do you have anything more to say about which summary statistic to use, or the potential implications of that choice? You use median instead of mean because of positively skewed distributions, but is it your general recommendation to use the median? Also see my point above about the second question (variability over time); the Discussion might be the place to address that issue.

Minor points:

In the results section, when talking about the measurements of exploration you state that no statistical tests were performed. It makes sense that these tests were unnecessary since they should be identical to the tests performed above, but it might be worth pointing that out for clarity. I admit to being confused as to why no tests were done at first.

It would be helpful for each graph in the figures to have a title. The information is in the caption, but it would be quicker and easier to understand if each panel had an informative title at the top.

In Figure 2A, what are the vertical bars? I assume the solid line is the median and the dashed on is the mean, but it would be nice to have that information clarified in the figure or caption

On page 9 you state “We approximated Δ by the median of trial-to-trial changes, instead of the mean” without explanation of why. It is explained later in the text, but it would help to have a quick explanation here since it is the first time it comes up (e.g. “due to extreme outliers” or “because the distributions were positively skewed”)

Typos etc:

On Pg. 7 in this sentence: “They were informed that after a familiarization phase, they would perform two phase phases”, the word phase is repeated

6. PLOS authors have the option to publish the peer review history of their article (what does this mean?). If published, this will include your full peer review and any attached files.

Reviewer #1: No

---

## [Author Response · Author response to Decision Letter 0]

14 Feb 2020

Response to the reviewers

Quantifying exploration in reward-based motor learning

PLOS ONE

Dear dr. Worthy,

Thank you for the constructive review. We feel that the reviewers’ comments have led to an improvement of the manuscript. We respond to the reviewers’ comments in detail below. We have also addressed the additional requirements that you mentioned in your decision letter: we have renamed the files according to the guidelines on your website, and we specified that participants provided informed written consent in the Methods section on page 5 of our manuscript. Furthermore, we slightly changed the figure colors to make them understandable for people who are colorblind and when printed in grayscale. 

Kind regards, also on behalf of my co-authors,

Nina van Mastrigt, MSc

 

Reviewer #1: In this study the authors systematically investigate ways of quantifying exploration in motor learning by separating it from variability due to sensorimotor noise. Using a target directed weight shift task, they designed an experiment wherein multiple different baselines for calculating variability due to exploration (which have all been variously used in the literature) can be used and compared. They conclude that using trial-to-trial change following rewarded trials provided a better baseline than trials in a baseline block or trials in no-feedback blocks. They also conclude (based on assessment of the uncertainty in the estimates) that the typical number of trials used may be insufficient for accurate estimations of variability due to exploration.

This is a nice study with a clever design that provides a valuable contribution that can help improve methods for future studies. The methods and analyses seem appropriate and sound. The conclusion seems justified and makes theoretical sense to me. Quantifying the uncertainty in the estimations was a nice touch. I think that it should be accepted for publication, but I think there are some issues the authors can address first. 

 Thanks for investing your time and energy in reading our manuscript. We are happy to hear that our conclusion makes theoretical sense to you, and that you consider the contribution of this study to the field valuable. 

How long did the whole experiment take participants approximately? This should be stated in the methods section somewhere. It is particularly important since you recommend an order of magnitude greater number of trials to be used in the future, and it would be good to have an idea of how long an experiment with that many trials would take. Would it be unreasonably long, or long enough that the participants’ motivation would become an issue?

 The experimental procedure lasted maximally 30 minutes. Performing all trials of the two experimental phases lasted only 13 minutes in total, but with intake, practice, and answering the Quick Motivation Index questions, the full experimental procedure took longer. We agree with you that the duration of the experiment should be stated in the methods section, and therefore added this information on page 7 and in more detail on page 8. In addition, based on one of your other comments on the number of trials in an experiment, we constructed a graph of confidence interval size for different trial numbers. We think that the reader could use this graph to determine the trade-off between motivation and number of trials needed for quantification of variability his- or herself, as motivation also depends on the experimental task. Thank you for this suggestion.

Here is one suggestion: you could expand on the bootstrapping analysis uses to quantify uncertainty by doing some simulations with larger numbers of trials and seeing how large the uncertainty of the estimate is. For example, there could be a graph of uncertainty as a function of number of trials. This could be really useful in determining the minimum number of trials to get a decent estimate. It might be that fewer additional trials are needed than your claim. At the other extreme, it’s possible that it asymptotes and even an unreasonably large number of trials gives a poor estimate. It would be cool (and potentially useful) to see what the curve of uncertainty by number of trials looks like.

 We really like this suggestion. We have added a graph of uncertainty of the estimates as a function of trial number, but only until 50-100 trials. We could not extend the number of trials to an amount higher than the number that we measured: with a higher number of trials, distributions of variability might change as a result of motivation (as you mentioned in your previous comment), fatigue or learning (as you mentioned in one of your next comments). We therefore sticked to simulating the estimation uncertainty for trial numbers up to 50-100. This resulted in figure 3A, which we replaced the original figures 3A&B with. Indeed, the confidence interval size depends on the number of trials, and decreases with higher numbers of trials.

Pg. 14 “No marked trends in variability over blocks were observed (Fig 3C).” I think that this point could use a little more discussion. I guess that it is good for the goals of the study that variability was constant across blocks, but shouldn’t you expect some decrease over time? Learning should decrease variability, and in general exploration should be highest early in a task when there is more to learn and should taper off later. Is it that most of the learning happens in the familiarization phase and beginning of the baseline block? Or is it that the stochastic feedback effectively prevents any real learning from occurring? A little discussion of these issues could be helpful. I think it makes sense that variability was stable, but my initial expectation was that it would go down over time.

 As we prevented participants from learning by providing them with stochastic feedback, indeed we did not expect a decrease in variability over time due to learning. As participants could not learn, we also did not expect higher exploration early in a task. This effect is probably driven by participants experiencing low success rates early in a task. In our task, the success rate was constant. Alternatively, you could have expected an increase in variability over time due to loss of motivation, but we found no significant difference in motivation between phases. We have added a paragraph in the discussion to clarify this for the readers. 

At the end of the introduction (pg. 4) you say “We also aimed to explore the statistical assessment of sensorimotor noise and exploration: which summary statistic should be used, how does variability change over time and how uncertain are sensorimotor noise and exploration estimates?” It would be nice to return to these questions a little bit more directly in the Discussion. The last question is discussed in detail, but the other two are only briefly touched on. Do you have anything more to say about which summary statistic to use, or the potential implications of that choice? You use median instead of mean because of positively skewed distributions, but is it your general recommendation to use the median? Also see my point above about the second question (variability over time); the Discussion might be the place to address that issue. 

 We agree with your comment. In the introduction, we state that the choice of summary statistics to describe variability could be a problem, especially when extreme values are present. These extreme values may either represent exploration or outliers. For this reason, using the median as a summary statistic provides a “safe” way to quantify the central tendency of the data. In our method, we quantify variability based on squared trial-to-trial changes, resulting in positive values by definition. This resulted in skewed distributions, whereas signed trial-to-trial changes might not have resulted in those distributions. We therefore recommend the use of the median as a summary statistic in our method. We have added this in the discussion. 

Minor points:

In the results section, when talking about the measurements of exploration you state that no statistical tests were performed. It makes sense that these tests were unnecessary since they should be identical to the tests performed above, but it might be worth pointing that out for clarity. I admit to being confused as to why no tests were done at first. 

 Thanks for pointing this out. We explicitly mentioned it now in the Statistics section on page 11, to make sure that readers will not get confused. We again clarified this in the Results section, on page 14. 

It would be helpful for each graph in the figures to have a title. The information is in the caption, but it would be quicker and easier to understand if each panel had an informative title at the top. 

 At first, we understood that this is not allowed by Plos One, but upon checking this a second time, we found out that we can indeed add a title to each graph. We agree with you that this will inform readers more efficiently about the content and meaning of each graph. We added titles now in all figures.

In Figure 2A, what are the vertical bars? I assume the solid line is the median and the dashed on is the mean, but it would be nice to have that information clarified in the figure or caption. 

 Thank you for discovering this omission! Your assumption was correct. We now clarified the meaning of the lines in the caption of figure 2A on page 13.

On page 9 you state “We approximated Δ by the median of trial-to-trial changes, instead of the mean” without explanation of why. It is explained later in the text, but it would help to have a quick explanation here since it is the first time it comes up (e.g. “due to extreme outliers” or “because the distributions were positively skewed”) 

 We agree with your idea to clarify this earlier. We therefore extended the sentence on page 9 with your latter suggestion. 

Typos etc:

On Pg. 7 in this sentence: “They were informed that after a familiarization phase, they would perform two phase phases”, the word phase is repeated. 

 We have removed the redundant word on page 7.

---

## [Decision Letter · Decision Letter 1]

17 Mar 2020

Quantifying exploration in reward-based motor learning

PONE-D-19-33458R1

Dear Dr. van Mastrigt,

We are pleased to inform you that your manuscript has been judged scientifically suitable for publication and will be formally accepted for publication once it complies with all outstanding technical requirements.

With kind regards,

Darrell A. Worthy, Ph.D

Academic Editor

PLOS ONE

Additional Editor Comments (optional):

Reviewers' comments:

Reviewer's Responses to Questions

**Comments to the Author**

1. If the authors have adequately addressed your comments raised in a previous round of review and you feel that this manuscript is now acceptable for publication, you may indicate that here to bypass the “Comments to the Author” section, enter your conflict of interest statement in the “Confidential to Editor” section, and submit your "Accept" recommendation.

Reviewer #1: All comments have been addressed

2. Is the manuscript technically sound, and do the data support the conclusions?

Reviewer #1: Yes

3. Has the statistical analysis been performed appropriately and rigorously? 

Reviewer #1: Yes

4. Have the authors made all data underlying the findings in their manuscript fully available?

Reviewer #1: Yes

5. Is the manuscript presented in an intelligible fashion and written in standard English?

Reviewer #1: Yes

6. Review Comments to the Author

Reviewer #1: The authors did a good job addressing all of my comments and suggestions, and I think that the paper is ready for publication. It is clearly written, the methods and analyses are sound, and the implications are well explained. The new versions of the figures look nice as well.

I noticed one small typo: at the top of pg. 4 the phrase "an estimate sensorimotor noise" seems to be missing an "of"

7. PLOS authors have the option to publish the peer review history of their article (what does this mean?). If published, this will include your full peer review and any attached files.

Reviewer #1: No

---

## [Editor Report · Acceptance letter]

20 Mar 2020

PONE-D-19-33458R1 

Quantifying exploration in reward-based motor learning 

Dear Dr. van Mastrigt:

I am pleased to inform you that your manuscript has been deemed suitable for publication in PLOS ONE. Congratulations! Your manuscript is now with our production department. 

With kind regards,

on behalf of

Dr. Darrell A. Worthy 

Academic Editor

PLOS ONE